# Retrospective analysis of neurofilament-light chain in patients with inflammatory bowel disease – A pilot study

Andreas Wolff[1]*, Emily Feneberg[1], Julius Shakhtour[2], Katja Steiger[2],
Roland M. Schmid[3], Bernhard Haller[4], Nya Reinhardt[1], Moritz Middelhoff[3],
David Schult-Hannemann[3], Paul Lingor[1,5,6]

**1** Clinical Department of Neurology, School of Medicine and Health, Technical University of Munich, Munich, Germany, **2** Department of Preclinical Medicine, Institute of Pathology, School of Medicine and Health, School of Medicine and Health, Technical University of Munich, Munich, Germany, **3** Department of Internal Medicine II, School of Medicine and Health, Technical University of Munich, Munich, Germany, **4** Institute for AI and Informatics in Medicine, School of Medicine and Health, Technical University of Munich, Munich, Germany, **5** German Center for Neurodegenerative Diseases (DZNE), Munich, Germany, **6** Munich Cluster of Systems Neurology (SyNergy), Munich, Germany

☉ These authors contributed equally to this work.
* andreas.wolff@tum.de

## Abstract

### Background

Chronic inflammatory bowel diseases, encompassing Crohn's disease and ulcerative colitis, are characterized by persistent inflammation of the gastrointestinal tract. While traditionally regarded as confined to the gut, the systemic nature of inflammatory bowel disease has been increasingly recognized. The nervous system has garnered particular attention due to molecular and clinical evidence suggesting a potential interplay between inflammatory bowel disease and neurodegenerative diseases. Inflammatory bowel disease patients have a higher risk of developing neurological disorders such as Parkinson's disease, all-cause dementia, and multiple sclerosis. Still, causative molecular mechanisms are poorly understood. Neurofilament light chain (NfL) has been established as a disease-independent biomarker of axonal damage reflecting neurodegeneration.

### Methods

In this pilot study, we assessed molecular evidence of neurodegeneration by measuring serum NfL in a single-molecule array using the HD-X SIMOA platform (Quanterix, MA, USA) and employing correlation with clinical data in forty-nine patients with histopathologically confirmed inflammatory bowel disease. In total, 24 Crohn's disease patients, 25 ulcerative colitis patients, and 23 controls, aged 18–79 years, were included.

**Data availability statement:** To protect participant privacy and comply with data protection regulations imposed by the responsible Ethics Committee individual-level data are made available to the public upon request. Requests for access to the research data should be directed to the Department of Neurology at TUM University hospital, Technical University of Munich (TUM), Munich, Germany (neurologie@mri.tum.de), where approval will be coordinated with the responsible Ethics Committee at TUM before any data are released.

**Funding:** Katja Steiger, Moritz Middelhoff, and David Schult-Hannemann received funding from the German Research Foundation/ Deutsche Forschungsgemeinschaft (DFG; CRC 1371, Project number 395357507). There was no additional external funding received for this study.

**Competing interests:** All authors declare that there are no competing interests in regard to this work.

**Abbreviations:** CD: Crohn's disease; UC: Ulcerative colitis; IBD: Inflammatory bowel disease; NfL: Neurofilament-light chain; CRP: C-reactive protein.

## Results

We found an age-dependency of serological NfL levels, however, no apparent differences between disease groups and controls. Crohn's disease patients showed a slower age-dependent incline in serological NfL compared to control subjects (p = 0.03). No correlation of NfL with disease duration, disease severity, or inflammatory bowel disease treatment was found.

## Conclusions

A slower age-dependent increase in serological NfL levels was found in Crohn's disease patients compared to control subjects. Larger studies assessing additional markers of neurodegeneration may be instrumental in addressing this question in the future.

## Introduction

Chronic inflammatory bowel disease (IBD), encompassing Crohn's disease and ulcerative colitis, are characterized by persistent inflammation of the gastrointestinal tract. These conditions involve complex interactions between genetic predisposition, immune dysregulation, environmental factors, and gut microbiota [1]. While traditionally regarded as confined to the gut, the systemic nature of IBD has been increasingly recognized, with its impact extending beyond the gastrointestinal tract to influence various extra-intestinal organs and systems. Among these, the central nervous system has garnered particular attention due to emerging evidence suggesting an interplay between IBD and neurological diseases [2,3]. Both being classified as immune-mediated inflammatory diseases, the co-occurrence of IBD and multiple sclerosis has been documented [4]. Microbiome dysbiosis, a hallmark of IBD, has been identified in multiple sclerosis patients, and genetic and environmental risk factors, such as vitamin D deficiency, smoking, and climate factors, are shared between these entities [5]. In large cohort analyses, chronic systemic inflammation, as assessed by increased levels of serum C-reactive protein (CRP), resulted in increased cognitive decline in a 20-year follow-up [6]. Additionally, the incidence of all-cause dementia was four times higher in patients with IBD while being diagnosed with dementia approximately 7 years earlier compared to controls [7]. While individual studies reported inconclusive results depending on sample size, ethnicity, and follow-up time, a recent meta-analysis of 5 cohort studies, including over 500,000 participants, found a pooled hazard ratio of 1.22 (95% confidence interval (CI): 1.05 to 1.38) between IBD and the risk of all-cause dementia [8]. In this meta-analysis, the most pronounced association was found for the risk of Parkinson's disease, with a hazard ratio of 1.39 (95% CI 1.20 to 1.58).

For numerous neurodegenerative diseases, the origin of the underlying pathology is currently unknown. However, increasing evidence suggests a close association between pathology in the gastrointestinal tract and the central nervous system, often preceding the diagnosis of a neurodegenerative disease by years. For example,

gastrointestinal symptoms such as constipation are frequently associated with the most common neurodegenerative movement disorder, Parkinson's disease, and an important hallmark of the disease, the aggregation of alpha-synuclein, can be detected early in the gastrointestinal tract [9], along with disturbances in the microbiome [10] and systemic inflammation [11–17]. Processes of gastrointestinal inflammation and microbiome disturbances are also evident in other neurodegenerative diseases, such as Alzheimer's disease and amyotrophic lateral sclerosis [18,19]. Conversely, anti-inflammatory therapy for IBD and prior intestinal denervation (vagotomy) are associated with a reduced risk of neurodegenerative diseases, specifically Parkinson's disease [20,21].

In recent years, neurofilament-light chain (NfL) has been widely studied as a non-specific marker of axonal and, therefore, neuronal damage in various neurological disorders, such as neurodegenerative diseases, multiple sclerosis, or stroke, and traumatic brain injury. While initially measured in cerebrospinal fluid, the development of single-molecule arrays enabled the assessment of NfL in the picomolar range in complex matrices, such as blood. Serum NfL has been established as a diagnostic marker, e.g., in the differential diagnosis of Parkinson's disease and atypical Parkinsonian disorders [22], as well as a prognostic marker, e.g., in the prediction of relapses in multiple sclerosis [23] or cognitive decline in Alzheimer's disease [24]. A growing body of evidence suggests that alterations in NfL levels can already be identified at pre-diagnostic stages. For example, a rise in NfL predicted the phenoconversion of pre-symptomatic gene mutation carriers for familial frontotemporal lobar degeneration, the second most common cause of dementia before the age of 60 [25], or familial amyotrophic lateral sclerosis up to 12 months before the development of first symptoms [26,27]. NfL was also found to be elevated in sporadic cases of amyotrophic lateral sclerosis up to 5 years prior to diagnosis. [28] As NfL is expressed not only in neurons of the central nervous system, peripheral neuropathies also result in elevated NfL levels [29]. Blood NfL levels are influenced by an complex interplay of release from peripheral neurons, migration over the blood-brain barrier from damaged central neurons, and elimination from the blood [30].

Taken together, NfL is considered the most promising biomarker for quantifying clinical and subclinical neurodegeneration independent of individual disease mechanisms. Its high sensitivity allows for the early detection of subtle neurodegenerative changes even before clinical symptoms appear, making it a valuable tool for risk assessment, disease monitoring, and evaluating potential neuroprotective interventions [26]. To date, no diagnostic tool is available to assess chronic neurodegenerative changes in patients with IBD. Therefore, measuring NfL in patients with IBD could be beneficial, as emerging evidence suggests a link between chronic systemic inflammation and an increased risk of neurodegeneration, potentially allowing for early intervention and monitoring in this at-risk population. To our knowledge, the potential role of NfL as a biomarker in IBD has not yet been explored [3]. In this pilot study, we assessed the utility of serum NfL for detecting signs of chronic neurodegeneration in a well-characterized cohort of patients with IBD at various disease stages, alongside matched healthy controls. Furthermore, we examined how clinical and paraclinical factors might influence NfL levels.

## Materials and methods

### Participants and data acquisition

Participants were prospectively recruited between December 2019 and January 2023 at the Klinikum rechts der Isar of the Technical University of Munich within the scope of the ColoBAC register study. Inclusion criteria were age ≥ 18 years and patients with gastrointestinal disease or suspicion thereof requiring endoscopic examination (colonoscopy and/or upper gastrointestinal endoscopy) or presenting for routine endoscopic follow-up or screening. Exclusion criteria included inability to consent, contraindications for biopsy (e.g., thrombocytopenia <50,000/µl, coagulation abnormalities), and poor general condition (ECOG >2 or Karnofsky <30%). Individuals received a colonoscopy with representative diagnostic biopsies. Classification into the IBD group was based on clinical, radiological, and histopathological parameters according to published guidelines [31]. Patients with no known IBD and no endoscopic or histologic signs of inflammation

were considered controls. The presence or absence of IBD was furthermore confirmed after a minimum of 12 months via phone call interviews. The absence of neurological comorbidities was confirmed by physical examination and by obtaining medical history. Age- and sex-matched patients subjects were selected randomly from the ColoBAC registry. Absence of neurological comorbidities and availability of serum samples was verified manually.

Approval was obtained from the ethics committee of the Technical University of Munich (No. 2018-322-7-S-SR). Written informed consent for colonoscopy and sample usage was obtained from all participants prior to the start of this study. The study was performed in accordance with good clinical practice and the Declaration of Helsinki. Sample analysis was additionally approved by the Use and Access committee and Ethics commission of the Technical University of Munich (No. 2024–73-S-SB). Data were accessed for research purposes on May 7, 2024.

### Data acquisition and classification

Clinical parameters, such as the use of immunosuppressants, were assessed using a questionnaire and the clinic's internal data system. The questionnaire also contained questions on lifestyle and demographic information (e.g., BMI, smoking, work, physical activity). Patients were considered to be physically active if at least one of the following criteria applied: a minimum of 2 hours of physical exercise per week, or at least 30 minutes of cycling per day, or at least 1 hour of walking per day. Disease severity was assessed using the Harvey Bradshaw Index for Crohn's disease [32] and the Mayo Score for ulcerative colitis [33], and categorized into remission, mild, moderate, and severe disease. Values of C-reactive protein (CRP) were measured as part of the clinical visit.

Biopsy samples were processed routinely and HE staining was performed according to standard protocols. Experienced pathologists performed histologic evaluation and diagnosis of IBD was made in accordance with valid recommendations [34] together with clinical information. Each biopsy site was assessed separately. Histologically, chronic inflammation was defined by a mononuclear inflammatory infiltrate. Active inflammation was present with an infiltration of neutrophil granulocytes, in the presence of crypt abscesses, or cryptitis. If there were features of both, chronic and active inflammation, the case was classified as active-chronic inflammation. Degree of inflammation was classified as low, moderate, or high grade, based on the density of the immune infiltrate. Likewise, crypt architectural disorder was classified in three grades (low, moderate, or high) based on the extent and severity of crypt atrophy, crypt shortening, and crypt branching. Each histopathological report was systematically evaluated, and the findings of the individual biopsy positions were summarized for each individual case. When multiple different grades of inflammation or crypt architectural disorder were present, for example low and high inflammatory activity or crypt architectural disorder in a patient, the highest grade per case was noted. In addition, The Nancy Index was employed to assess and grade the histological severity of inflammation. The Nancy Index uses a five-level scale from 0 to 4, with 0 indicating no relevant histological disease activity and 1–4 representing increasing degrees of inflammation and tissue damage, up to grade 4, which reflects severe activity such as ulcers or erosions. The Nancy Index was always based on the highest degree of inflammation [35].

### Sample preparation and NfL measurement

Serum samples were collected, centrifuged at 2,000 rpm at room temperature for 10 min and frozen immediately at −20°C. For longtime storage (>1 month) samples were stored at −80°C. Samples were processed as singlets in a single batch on a fully automated SIMOA HD-X Analyzer (Quanterix, Lexington, MA, USA) using the NF-LIGHT™ Neurofilament Light Chain Assay (NF-Light v2 Advantage, Lot Nr. 503892). The investigators and analysts were blinded to the diagnosis. The analytic lower limit of quantification for this assay is 0.345 pg/mL (coefficient of variance (CV) 18.9%). Commercial standards and in-house control were within the expected ranges (intra-assay CV 3.5% and 3.0%, respectively). One sample was excluded due to debris detected by the analyzer.

 

## Statistical analysis

All statistical calculations were carried out using R Version 4.1.0 (R Core Team, Vienna, Austria). Because NfL values were heavily skewed, natural logarithm transformation was used to help achieve normality. Welch two sample t-tests (for metric data), Fisher exact tests (for categorical data), and a permutation test for trends (from *coin* package for ordinal data) were performed for comparison of variables between two groups, whereas linear models (ANOVA/ANCOVA, for metric data) and Pearson's Chi-square test (for categorical data) were performed for comparison of three groups. As a non-parametric test, a Kruskal-Wallis test with unadjusted post-hoc pairwise Wilcoxon test was performed. The correlation between two metric variables was assessed by Pearson's product-moment correlation. Linear models included age as an independent variable, if not stated otherwise, and regression coefficients, as well as 95% confidence intervals are reported. If categorical variables included more than two levels, the global effect of the variable was assessed by ANOVA. Age-matching was achieved using the *matchit* function from the package *MatchIt* (Version 4.5.5) and paired-pairwise t-tests were calculated [36]. Detailed test statistics are provided in the Supplementary Material.

## Results

### Participants

This study included 49 IBD patients (24 patients with Crohn's disease and 25 patients with ulcerative colitis), and 23 age- and sex-matched controls. All groups showed comparable demographics (Table 1).

### Serum NfL in patients with inflammatory bowel disease and controls

One patient with ulcerative colitis was excluded due to debris in the sample. Overall, all subjects displayed serum NfL values above the limit of detection. Because NfL values were heavily skewed, natural logarithm transformation was

**Table 1. Demographics and clinical characteristics of study participants.**

| Parameter | Control | Crohn's disease | Ulcerative colitis | P-value |
|---|---|---|---|---|
| N | 23 | 24 | 25 | |
| Age, years, mean (range) | 45.2 (23-79) | 39.8 (18-74) | 43.9 (18-76) | 0.44[A] |
| Sex, female | 11 (48%) | 12 (50%) | 9 (36%) | 0.57[C] |
| BMI, kg/m$^2$, mean (SD) | 24.8 (4.1) | 24.1 (4.5) | 23.1 (4.5) | 0.40[A] |
| Current tobacco consumer | 4 (17%) | 4 (17%; missing = 1) | 2 (8%) | 0.43[C] |
| Physically active[1] | 19 (83%) | 19 (79%; missing = 1) | 14 (56%) | 0.10[C] |
| Country of birth: Germany | 20 (87%) | 20 (83%; missing = 2) | 21 (84.0%; missing = 1) | 0.45[C] |
| Age at initial diagnosis, years, mean (range) | – | 27.3 (9-74) | 29.6 (17-61) | 0.50[A*] |
| Disease duration, years, mean (range) | – | 12.6 (−1-44) | 14.4(0-52) | 0.64[A*] |
| Disease activity | | | | |
| (1) Remission<br> (2) Mild<br> (3) Moderate<br> (4) Severe | – | 9 (43%)<br>4 (19%)<br>7 (33%)<br>1 (5%) | 6 (25%)<br>3 (13%)<br>13 (54%)<br>2 (8%) | 0.13[T*] |
| Immunosuppressive therapy present | 1 (4%)[2] | 15 (63%) | 24 (96%) | <0.001[C]; 0.005[F*] |
| Rectal or oral steroids present | 1 (4%)[2] | 0 | 1 (4%) | 1[F*] |
| IV steroids present | 0 | 1 (4%) | 0 | 0.49[F*] |
| Biologicals present | 0 | 9 (38%) | 9 (36%) | 1[F*] |

SD standard deviation, BMI body mass index, CRP C reactive protein, 1 "Physically active" was defined as minimum of 2 hours of physical exercise per week, or at least 30 minutes of cycling per day, or at least 1 hour of walking per day, 2 not IBD-related reason, A Linear model ANOVA, C Pearson's Chi-square test, T Trend test for ordinal variables, F Fisher exact test, *comparison between Crohn's disease and ulcerative colitis.

used to help achieve normality. Serum NfL levels showed a strong age-dependency (Pearson's r = 0.66, p < 0.001, Fig 1A); we found no significant dependency of NfL from sex (Welch two-sample t-test; p = 0.81) or BMI (Pearson's r = 0.14, p = 0.26). NfL values were not significantly different between the groups: controls (18.3 ± 19.8 pg/mL), Crohn's disease (12.6 ± 17.3 pg/mL), and ulcerative colitis (14.2 ± 16.6 pg/mL, Fig 1B). Three different statistical methods assessed differences between groups. Neither in a linear model (p = 0.53) nor an age-adjusted linear regression model (global ANOVA: p = 0.71, comparison to control: Crohn's disease: p = 0.70, UC: p = 0.42) nor an age-matched paired pairwise t-tests (CO vs. CD: p = 0.19, CO vs. UC: p = 0.14, CD vs. UC: p = 0.72) identified significant group differences. Additionally, we examined the interaction between age and group (age*group) on the outcome variable NfL in a linear regression model. Here, NfL change per year of age (interaction term) significantly differed between Crohn's disease patients and control subjects (regression coefficient ($\beta_{int}$): −0.03 (95% CI: −0.05 to 0.00), p = 0.031), while no significant difference was found between ulcerative colitis patients and controls subjects ($\beta_{int}$: −0.01 (95% CI: −0.03 to 0.01), p = 0.38).

### Correlation of NfL with disease activity and treatment

Systemic inflammation was assessed by levels of C-reactive protein (CRP). Here, disease groups (CD: 1.28 (±2.7) mg/dl; UC: 1.04 (±2.6) mg/dl) showed higher CRP levels than controls (CO: 0.61 (±2.2) mg/dl; Kruskal-Wallis test: p = 0.004, unadjusted post-hoc pairwise Wilcoxon test: CO vs. CD: p = 0.002, CO vs. UC: p = 0.027, CD vs. UC: p = 0.15). However, CRP levels (log-transformed) were not associated significantly with serum NfL levels (age-adjusted regression coefficient (β): 0.09 (95% CI: −0.01 to 0.19), p = 0.07). As presented in Table 2, multiple clinical and histological outcome parameters were assessed for their association with altered NfL values, however, no significant association was identified. While macroscopic inflammation during colonoscopy was different between groups (CO: 4%, CD: 74%, UC: 84%, Pearson's Chi-square test: p < 0.001; unadjusted post-hoc Fisher's exact test: CO vs. CD: p < 0.001, CO vs. UC: p < 0.001, CD vs. UC: p = 0.32), it was not associated significantly with NfL. Overall, we found disease duration to be negatively associated with disease activity (age-adjusted global ANOVA: p = 0.013). This association was additionally identified in patients with

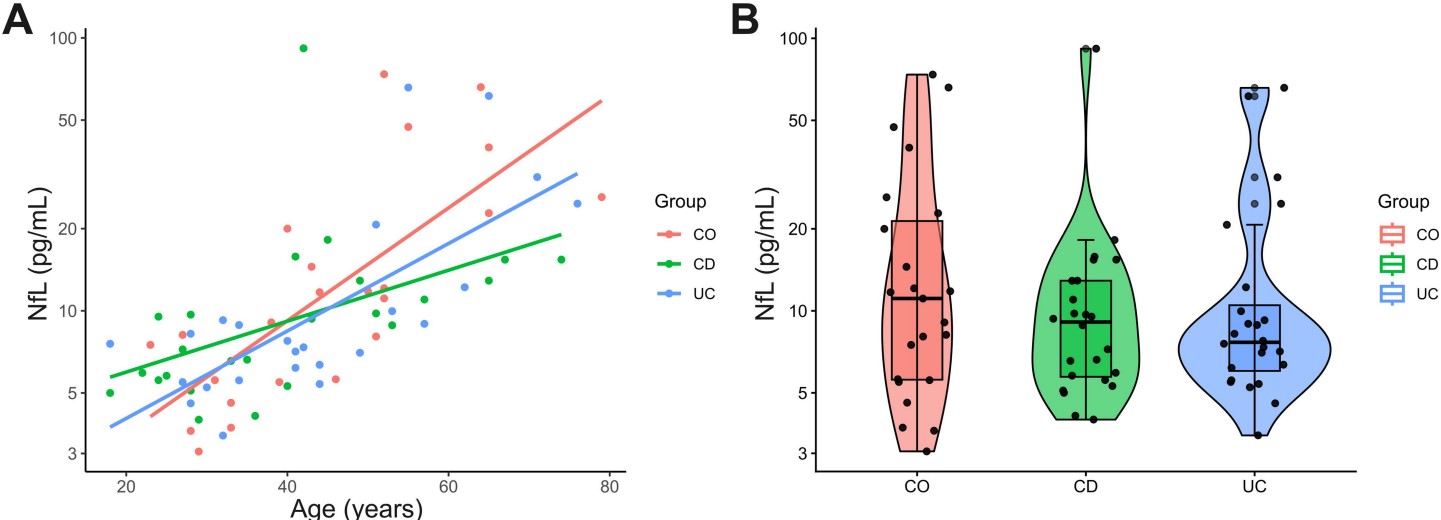

**Fig 1. A Correlation analysis of age and NfL for patients with Crohn's disease (CD), ulcerative colitis (UC), and controls (CO).** The Y-axis represents absolute NfL concentrations on a natural logarithmic scale. **B** Serum NfL in patients with and without a diagnosis of chronic inflammatory bowel disease (CD N = 24, UC N = 25) and controls (N = 23). The boxes map to the median, 25th, and 75th percentiles, Whiskers extend to the range of values within Q3 + 1.5 × IQR to Q1 - 1.5 × IQR, p = 0.53 according to ANOVA. The Y-axis represents absolute NfL concentrations on a natural logarithmic scale.

**Table 2. Association of NfL with clinical and histological parameters of inflammatory bowel disease patients and control subjects.**

| Clinical and histological outcome parameter | Age-adjusted regression coefficient (β) (95% CI) | P (age-adjusted) |
|---|---|---|
| Macroscopic inflammation during colonoscopy (yes/no) | −0.08 (−0.37 to 0.21) | 0.58 |
| Histological subclassification of activity[1] | n.a. | 0.71[g] |
| Histological disease activity (Nancy index)[2] | n.a. | 0.26[g] |
| Distribution pattern of intestinal inflammation[3] | n.a. | 0.69[g] |
| Presence of architectural distortion of the colonic mucosa (yes/no) | −0.01(−0.32 to 0.30) | 0.96 |
| Degree of the architectural distortion of the colonic mucosa[4] | n.a. | 0.98[g] |
| Clinical disease activity[5] | n.a. | 0.26[g] |
| Disease duration (years) | −0.00 (−0.02 to 0.01) | 0.66 |
| Age at initial IBD diagnosis (years) | 0.00 (−0.01 to 0.02) | 0.66 |

For each outcome parameter an age-adjusted linear model was fitted. If categorical variables included more than two levels, the global effect of the variable was assessed by ANOVA. In these cases, not available (n.a.) is stated for β and the P value is marked with g (global ANVOA); 1 none, active, active-chronic, chronic; 2 ranging from 0 (no histological significant disease) to 4 (severely active disease); 3 no inflammation, focal, or diffuse; 4 ranging from none, mild, moderate to severe; 5 ranging from remission to mild, moderate, to severe; IBD inflammatory bowel disease

Crohn's disease (age-adjusted global ANVOA: p = 0.01), however, not in ulcerative colitis patients (age-adjusted global ANOVA: p = 0.55). However, no apparent association between NfL and disease activity was observed. Also on the group level, no significant association between NfL and disease activity was also found (age-adjusted global ANOVA: CD patients: p = 0.24, UC patients: p = 0.51). Disease duration correlated strongly with age (Pearson's r = 0.65, p < 0.001), however, no significant influence on NfL was identified. We additionally assessed current IBD-specific treatment on its effect on NfL. Here, neither presence of treatment (age-adjusted β: −0.02 (95% CI: 0.45 to 0.41), p = 0.91), nor oral/topic steroids (age-adjusted β: −0.14 (95% CI: −1.32 to 1.03), p = 0.81), or IV steroids (age-adjusted β: −0.23 (95% CI: −1.41 to 0.94), p = 0.69) were associated with NfL. Treatment with biologicals was also not associated with altered NfL values (age-adjusted β: 0.10 (95% CI: −0.25 to 0.45), p = 0.58).

## Discussion

To the best of our knowledge, this is the first study assessing NfL as a marker of subclinical neuronal damage in patients with IBD. [3] A state-of-the-art single-molecule immuno-assay technique was applied, enabling NfL measurements in the picomolar range in blood. The cohort consisted of 49 well-characterized patients with IBD (24 Crohn's disease and 25 ulcerative colitis cases) and 23 control subjects.

### No evidence for altered serological NfL levels in inflammatory bowel disease patients

In line with previous reports, we found a strong age-dependency of NfL [37]. Neither age-adjustment nor age-matched paired-pairwise testing identified significant differences in NfL levels between the disease entities and controls. However, the age-dependent increase of NfL in patients with Crohn's disease was significantly lower than in control cases. This suggests that younger Crohn's disease patients with a shorter disease duration show higher NfL values, and older Crohn's disease patients with a longer disease duration show lower NfL values than controls. Given that younger age at diagnosis is a known predictor for more aggressive disease progression in CD, this observation may reflect neuroaxonal injury occurring early in the disease course, which stabilizes or declines as disease activity subsides over time [38]. The

negative correlation between disease activity and disease duration in CD patients further supports this interpretation. In consideration of the small sample size, the absence of an independent association between disease activity and NfL values may also reflect measurement sensitivity, or the complex relationship between systemic inflammation and neuro-axonal injury. While elevated systemic inflammation, as reflected by C-reactive protein, was observed in IBD groups, we found no direct correlation between CRP levels with serum NfL levels. Similarly, present administration of anti-inflammatory therapies, including corticosteroids and biological agents, did not significantly influence serum NfL concentrations. However, the effect of long-term anti-inflammatory treatment for IBD on the development of neurological diseases is unclear. Epidemiological studies suggest that early IBD treatment with anti-TNF therapy resulted in a 78% reduced Parkinson's disease incidence rate compared to IBD patients not exposed [39]. Similar results were found for azathioprine [40]. Therefore, past long-term anti-inflammatory therapy may contribute to the observed slower age-dependent increase in serum NfL levels in Crohn's patients. Since no significant changes were observed in patients with ulcerative colitis, this underscores the distinct pathophysiological characteristics of IBD, not only in their intestinal manifestations but also in their potential neurological involvement. Despite no overall difference in NfL between groups, the altered age-NfL relationship in CD highlights a potentially unique neurodegenerative pattern in a subgroup of patients. Further investigation is warranted to assess whether NfL might serve as a marker for early neurodegeneration, especially in younger patients with aggressive disease phenotypes. Identifying these patient subgroups is of utmost importance, as it enables targeted intensive treatment strategies, ideally employing therapeutics demonstrated to have neuroprotective properties. Early identification and intervention in these individuals could substantially mitigate the long-term risk and burden of neurode-generative comorbidities in IBD patients.

## Normal NfL levels do not exclude neurodegeneration

Smoldering neurodegeneration can result in elevated NfL values on the group level, as it has been shown in presymp-tomatic amyotrophic lateral sclerosis patients [26,27]. It is important to note that normal serum NfL concentrations do not preclude the presence of neurodegenerative changes. NfL reflects ongoing axonal damage dynamics, thus slow or low-grade neurodegeneration might not yield measurable elevations. This is consistent with observations in early or slowly progressive neurodegenerative diseases such as Parkinson's disease, where NfL levels can remain near nor-mal despite pathology [30,41], especially in the early stages of the disease. Imaging studies have reported structural brain changes in IBD patients, supporting the existence of subclinical neurodegeneration not captured by peripheral NfL measurements alone [42]. Moreover, our assessment was confined to serum measurements. Although serum and cerebrospinal fluid (CSF) NfL levels correlate well, CSF NfL remains the more sensitive biomarker for neurodegenera-tion [43]. Future studies integrating CSF analyses with advanced neuroimaging and cognitive assessments will be crit-ical to comprehensively characterize neuronal injury in IBD and validate serum NfL's utility as a peripheral biomarker in this context.

## Limitations

Our study has several limitations: IBD patients of various disease severity and disease duration were included in this study, making this cohort demographically representative of a majority of IBD cases, but limiting the sample sizes for the individual disease stage. However, studies assessing NfL in presymptomatic mutation carriers for neurological diseases reported altered NfL levels in small cohorts of comparable size [26,28]. While this study provides evidence for age-depended changes of NfL in patients with Crohn's disease, larger sample sizes will be essential for robust validation. Furthermore, no long-term follow-up data or thorough neuropsychological testing was available. Therefore, neither mild cognitive nor neurological alterations could be evaluated, and no data on the future development of neurocognitive deficits or neurological symptoms can be provided.

## Conclusions

In this pilot study, serum NfL levels in IBD patients did not significantly differ from control subjects without intestinal inflammation. However, Crohn's disease patients exhibited a slower age-dependent increase in serum NfL levels compared to controls, which may be attributable to the effects of long-term anti-inflammatory treatments or a reduction in disease activity as patients age. Prospective longitudinal studies incorporating multi-modal biomarker panels, neuroimaging, and cognitive testing in diverse IBD populations, particularly younger individuals with aggressive disease, are imperative. These approaches will clarify the pathophysiological relevance of neuroaxonal injury in IBD and might identify patients at risk for neurological comorbidities, opening avenues for early intervention.

## Supporting information

**S1 File. Detailed test statistics.**
(DOCX)

## Acknowledgments

The authors are deeply indebted to the volunteer participants and all those involved in the design and execution of this trial. We thank Christina Gassner and Niklas Thur for help with patient data collection. We thank the tissue biobank of Klinikum rechts der Isar and TUM (IBioTUM) namely, Klaus-Peter Janssen and Anja Conrad, for excellent technical support.

## Author contributions

**Conceptualization:** Andreas Wolff, Moritz Middelhoff, David Schult-Hannemann, Paul Lingor.

**Data curation:** Andreas Wolff, Julius Shakhtour, Katja Steiger, Nya Reinhardt, David Schult-Hannemann.

**Formal analysis:** Andreas Wolff, Bernhard Haller, Paul Lingor.

**Funding acquisition:** Emily Feneberg, Katja Steiger, Roland M. Schmid, Moritz Middelhoff, David Schult-Hannemann, Paul Lingor.

**Investigation:** Andreas Wolff, Emily Feneberg, Julius Shakhtour, Nya Reinhardt, David Schult-Hannemann.

**Methodology:** Andreas Wolff, Moritz Middelhoff, Paul Lingor.

**Project administration:** Andreas Wolff, David Schult-Hannemann.

**Resources:** Roland M. Schmid, Paul Lingor.

**Software:** Bernhard Haller.

**Supervision:** Katja Steiger, Roland M. Schmid, Bernhard Haller, Moritz Middelhoff, Paul Lingor.

**Validation:** Andreas Wolff, Emily Feneberg, Bernhard Haller, David Schult-Hannemann, Paul Lingor.

**Visualization:** Andreas Wolff, Bernhard Haller, Moritz Middelhoff.

**Writing – original draft:** Andreas Wolff.

**Writing – review & editing:** Emily Feneberg, Julius Shakhtour, Katja Steiger, Roland M. Schmid, Bernhard Haller, Nya Reinhardt, Moritz Middelhoff, David Schult-Hannemann, Paul Lingor.

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
