## [Decision Letter · Decision Letter 0]

26 Aug 2025

Dear Dr. Andreas W. Wolff

Thank you for submitting your manuscript to PLOS ONE. After careful consideration, we feel that it has merit but does not fully meet PLOS ONE’s publication criteria as it currently stands. Therefore, we invite you to submit a revised version of the manuscript that addresses the points raised during the review process.

We look forward to receiving your revised manuscript.

Kind regards,

Kazuo Sugimoto

Academic Editor

PLOS ONE

Journal Requirements:

“KS, MM, and DSH received funding from the German Research Foundation (DFG, CRC1371, Project number 395357507, https://www.dfg.de/en).”

4. In the online submission form you indicate that your data is not available for proprietary reasons and have provided a contact point for accessing this data. Please note that your current contact point is a co-author on this manuscript. According to our Data Policy, the contact point must not be an author on the manuscript and must be an institutional contact, ideally not an individual. Please revise your data statement to a non-author institutional point of contact, such as a data access or ethics committee, and send this to us via return email. Please also include contact information for the third party organization, and please include the full citation of where the data can be found.

5. Thank you for stating the following in your manuscript:

“Katja Steiger, Moritz Middelhoff, and David Schult-Hannemann received funding from the German Research Foundation (DFG, CRC1371, Project number 395357507).”

“KS, MM, and DSH received funding from the German Research Foundation (DFG, CRC1371, Project number 395357507, https://www.dfg.de/en).”

Additional Editor Comments (if provided):

Please make point-by-point revisions according to the reviewers' comments and suggestions.

Reviewers' comments:

Reviewer's Responses to Questions

**Comments to the Author**

1. Is the manuscript technically sound, and do the data support the conclusions?

Reviewer #1: Partly

2. Has the statistical analysis been performed appropriately and rigorously?

Reviewer #1: Yes

3. Have the authors made all data underlying the findings in their manuscript fully available?

Reviewer #1: Yes

4. Is the manuscript presented in an intelligible fashion and written in standard English?

Reviewer #1: Yes

Reviewer #1: The Introduction is quite elaborate, but the most important information are missing i.e. what is known until now about the role of neurofilament-light chain in inflammatory bowel disease, what is the knowledge gap, and why it this area should be explored. Wnhat this study would add to the current knowledge?

Please define clear aim of the study.

The most significant concern refers to methodology.

The study lacks a calculation of the sample size which implies a question about the power of the study to provide statistically and clinically relevant results.

The authors also neglect to provide eligibility criteria for the study and control group. Why one patient from healthy control group was on immunosupressive therapy?

Why you did not performed any neurocognitive test?

Clinical characteristics of patients with IBD is missing (phenotype, location, activity, age at onset, co-morbidities etc.)

The discussion does not provide explanation of results, nor any significant implications

**Do you want your identity to be public for this peer review?** For information about this choice, including consent withdrawal, please see our Privacy Policy

Reviewer #1: No

---

## [Author Response · Author response to Decision Letter 1]

2 Sep 2025

Dear Editors, dear Reviewer,

Thank you very much for your kind suggestions.

In the following we respond to your comments point by point. Citations of our initial manuscript version are formatted in italics. Changes to the original text are given in italics and underlined.

Editor comments:

Question 1: 1. Please ensure that your manuscript meets PLOS ONE's style requirements, including those for file naming. The PLOS ONE style templates can be found at….

Reply 1: We changed manuscript style, and it now meets the PLOS ONE’s style requirements.

Question 2: 2. Thank you for stating in your Funding Statement:

“KS, MM, and DSH received funding from the German Research Foundation (DFG, CRC1371, Project number 395357507, https://www.dfg.de/en).”

Reply 2: Thank you for this input, we adapted funding statement and cover letter accordingly.

Question 3: 3. We note that you have indicated that there are restrictions to data sharing for this study. For studies involving human research participant data or other sensitive data, we encourage authors to share de-identified or anonymized data. However, when data cannot be publicly shared for ethical reasons, we allow authors to make their data sets available upon request. For information on unacceptable data access restrictions, please see http://journals.plos.org/plosone/s/data-availability#loc-unacceptable-data-access-restrictions.

Reply 3: We added the following to the Data availability statement: To protect participant privacy and comply with data protection regulations imposed by the responsible Ethics Committee individual-level data are made available to the public upon request. Requests for access to the research data should be directed to the Department of Neurology at TUM University hospital, Technical University of Munich (TUM), Munich, Germany (neurologie@mri.tum.de), where approval will be coordinated with the responsible Ethics Committee at TUM before any data are released.

Question 4: 4. In the online submission form you indicate that your data is not available for proprietary reasons and have provided a contact point for accessing this data. Please note that your current contact point is a co-author on this manuscript. According to our Data Policy, the contact point must not be an author on the manuscript and must be an institutional contact, ideally not an individual. Please revise your data statement to a non-author institutional point of contact, such as a data access or ethics committee, and send this to us via return email. Please also include contact information for the third party organization, and please include the full citation of where the data can be found.

Reply 4: Please see Reply 3.

Question 5: 5. Thank you for stating the following in your manuscript:

“Katja Steiger, Moritz Middelhoff, and David Schult-Hannemann received funding from the German Research Foundation (DFG, CRC1371, Project number 395357507).”

“KS, MM, and DSH received funding from the German Research Foundation (DFG, CRC1371, Project number 395357507, https://www.dfg.de/en).”

Reply 5: We apologize for the confusion and would like to clarify this: The Acknowledgement section currently states: We thank the tissue biobank of Klinikum rechts der Isar and TUM (IBioTUM) namely, Klaus-Peter Janssen and Anja Conrad, for excellent technical support.

This refers to the tissue biobank funded through German Research Foundation (DFG, CRC1371, Project number 395357507, https://www.dfg.de/en) and is not related to additional external funding. Therefore, we ask to remain with the original statement.

Question 6: 6. Your ethics statement should only appear in the Methods section of your manuscript. If your ethics statement is written in any section besides the Methods, please delete it from any other section.

Reply 6: We removed the ethics statement from the supplementary statements at the end of the manuscript. It now only appears in the Methods section of the manuscript.

Question 7: 7. If the reviewer comments include a recommendation to cite specific previously published works, please review and evaluate these publications to determine whether they are relevant and should be cited. There is no requirement to cite these works unless the editor has indicated otherwise.

Reply 7: Not applicable.

Reviewers' comments:

Reviewer #1:

Question 1: The Introduction is quite elaborate, but the most important information are missing i.e. what is known until now about the role of neurofilament-light chain in inflammatory bowel disease, what is the knowledge gap, and why it this area should be explored. Wnhat this study would add to the current knowledge? Please define clear aim of the study.

Reply 1: We thank the reviewer for the important remark. We adapted the introduction:

Taken together, NfL is considered the most promising biomarker for quantifying clinical and subclinical neurodegeneration independent of individual disease mechanisms. Its high sensitivity allows for the early detection of subtle neurodegenerative changes even before clinical symptoms appear, making it a valuable tool for risk assessment, disease monitoring, and evaluating potential neuroprotective interventions.(26) To date, no diagnostic tool is available to assess chronic neurodegenerative changes in patients with inflammatory bowel diseases. Therefore, measuring NfL in patients with inflammatory bowel disease could be beneficial, as emerging evidence suggests a link between chronic systemic inflammation and an increased risk of neurodegeneration, potentially allowing for early intervention and monitoring in this at-risk population. To our knowledge, the potential role of NfL as a biomarker in inflammatory bowel disease has not yet been explored. (3) In this pilot study, we assessed the utility of serum NfL for detecting signs of chronic neurodegeneration in a well-characterized cohort of patients with inflammatory bowel disease at various disease stages, alongside matched healthy controls. Furthermore, we examined how clinical and paraclinical factors might influence NfL levels.

Question 2: The most significant concern refers to methodology.

The study lacks a calculation of the sample size which implies a question about the power of the study to provide statistically and clinically relevant results.

Reply 2: This is an important remark, and we acknowledge this limitation. Our study was conducted using a historical cohort of patients with inflammatory bowel disease and matched controls; furthermore, our study is the first to assess NfL levels in IBD patients, therefore no statistics were available to estimate the sample size. Therefore, an a-priori sample size calculation was not performed. We have addressed this limitation explicitly in the discussion section of the manuscript.

We would like to emphasize that previous studies with comparable sample sizes have successfully detected significant changes in NfL levels, even in relatively small patient cohorts. Since this is the first study evaluating NfL in patients with inflammatory bowel disease, no prior estimates of mean values or standard deviations are available for this population.

To provide context, the largest body of evidence for NfL comes from studies on dementia patients, where a mean serum NfL concentration of 19 ± 12 pg/mL has been reported for patients with Alzheimer’s disease, compared to 10 ± 7 pg/mL in healthy controls (PMID: 36196979). Based on these values, a sample size of approximately 23 subjects per group would be required to detect differences using a two-sided t-test with 80% power and a significance level of 0.05. Therefore, we consider the sample size in our study sufficient to detect clinically meaningful differences, although we agree that validation in larger cohorts is warranted.

Question 3: The authors also neglect to provide eligibility criteria for the study and control group.

Reply 3: We additionally added information on the selection process for this study to the methods section: Inclusion criteria were age ≥18 years and patients with gastrointestinal disease or suspicion thereof requiring endoscopic examination (colonoscopy and/or upper gastrointestinal endoscopy) or presenting for routine endoscopic follow-up or screening. Exclusion criteria included inability to consent, contraindications for biopsy (e.g., thrombocytopenia <50,000/µl, coagulation abnormalities), and poor general condition (ECOG >2 or Karnofsky <30%). […] Age- and sex-matched subjects were selected randomly from the ColoBAC registry. Absence of neurological comorbidities and availability of serum samples was verified manually.

Question 4: Why one patient from healthy control group was on immunosupressive therapy?

Reply 4: Thank you for raising this point. We have clarified this detail in Table 1, now indicating that the immunosuppressive treatment administered to this patient was for a renal condition, not related to inflammatory bowel disease. During colonoscopy, we confirmed the absence of intestinal inflammation in this patient, and a thorough review of the medical records verified that the patient had no neurological disorders or relevant comorbidities expected to confound NfL levels.

Question 5: Why you did not performed any neurocognitive test?

Reply 5: As this pilot study was based on an existing historical cohort of IBD patients, the original study design did not include assessment of neurocognitive function. Given the considerable time elapsed between the baseline sampling and the current follow-up period, we determined that it would not be appropriate to evaluate neurocognitive function at this stage. However, we thank the reviewer for this important question, and this limitation is specifically addressed in the relevant section of the manuscript.

Question 6: Clinical characteristics of patients with IBD is missing (phenotype, location, activity, age at onset, co-morbidities etc.)

Reply 6: Thank you for your comment. We have added important additional clinical characteristics to Table 1, which now includes age at initial diagnosis, disease duration, and disease activity. We chose not to include all available clinical data—for example, detailed comorbidities—due to the complexity and heterogeneity of these variables. However, as stated in the methods section, patients and control subjects with neurological comorbidities known to affect NfL levels were excluded from the study to minimize confounding.

Question 7: The discussion does not provide explanation of results, nor any significant implications

Reply 7: Thank you very much for your remark. In response, we have revised the manuscript to strengthen the discussion regarding the observed slower age-dependent increase in serum NfL levels among patients with Crohn’s disease. We have more explicitly elaborated on potential reasons that might underly this finding.

Moreover, we have placed additional emphasis on the necessity for prospective longitudinal studies that incorporate multi-modal biomarker panels, advanced neuroimaging techniques, and comprehensive cognitive assessments across diverse populations of patients with inflammatory bowel disease. Such studies are essential to unravel the complex processes involved in neuronal injury and neurodegeneration in this patient population.

Beyond revisions in wording, we also added clear implications to guide future research aimed at identifying clinically relevant subgroups of IBD patients who may present with biomarker evidence of neurodegeneration. This stratification could be crucial for earlier detection and potential therapeutic interventions targeting neurological comorbidities in inflammatory bowel disease.

6. PLOS authors have the option to publish the peer review history of their article (what does this mean?). If published, this will include your full peer review and any attached files.

Do you want your identity to be public for this peer review? For information about this choice, including consent withdrawal, please see our Privacy Policy.

Reviewer #1: No

---

## [Decision Letter · Decision Letter 1]

1 Dec 2025

Dear Dr. Wolff,

Thank you for submitting your manuscript to PLOS ONE. After careful consideration, we feel that it has merit but does not fully meet PLOS ONE’s publication criteria as it currently stands. Therefore, we invite you to submit a revised version of the manuscript that addresses the points raised during the review process.

We look forward to receiving your revised manuscript.

Kind regards,

Andrea Calcagno

Academic Editor

PLOS ONE

Journal Requirements:

**Additional Editor Comments:**

Thanks for addressing most of the reviewers' concerns.

They still have a few minor comments before we can quickly proceed.

Reviewers' comments:

Reviewer's Responses to Questions

**Comments to the Author**

Reviewer #2: (No Response)

Reviewer #3: (No Response)

2. Is the manuscript technically sound, and do the data support the conclusions?

Reviewer #2: Yes

Reviewer #3: Partly

3. Has the statistical analysis been performed appropriately and rigorously?

Reviewer #2: Yes

Reviewer #3: Yes

4. Have the authors made all data underlying the findings in their manuscript fully available?

Reviewer #2: Yes

Reviewer #3: Yes

5. Is the manuscript presented in an intelligible fashion and written in standard English?

Reviewer #2: Yes

Reviewer #3: Yes

Reviewer #2: The authors provided a thorough revision addressing the reviewers' main concerns. Nevertheless, lack of neurocognitive testing and the relatively small sample size without an a priori power calculation is still an issue. While these are acknowledged in the discussion, they limit the strength of conclusions. Yet, the study is overall technically sound, the data support the main conclusion that serum neurofilament-light chain (NfL) levels show age-dependency but no significant difference between inflammatory bowel disease (IBD) patients and controls, although Crohn's disease showed a distinct age-dependent pattern. The conclusions are appropriately cautious given the limitations.

Minor points:

1) I recommend to emphasize the pilot nature of this study clearly and suggest future larger, longitudinal studies with cognitive testing and multimodal biomarkers.

2) I suggest adding a Supplementary Information showing other statistical values besides p value such as t and df values for t-test, df, P value for ANOVA, H and df for Kruskal Wallis etc. Violin box plot showing single data points shall be provided and max/min should show the horizontal termination bar (similar to STD error bars, see also https://www.nature.com/articles/s41598-023-29704-8/figures/1). In addition. wildcards should show significance directly in graphs (Fig. 1B).

3) The author sequence is a bit unusual. It says “equal contribution” for the two last authors though this is usually done for first authors. Also, the last author is commonly the PI receiving funds which is here not the case. I suggest to clarify this.

Reviewer #3: (1) In my view, the authors have adequately addressed the concerns of the prior reviewer. Overall, the manuscript is well constructed, and the conclusions largely are justified by the experimental data presented. I have two minor comments for the authors for their consideration:

(2) Methods. Data Acquisition and Classification. This section (with regard to histological findings) is confusing and could be structured for clearer readability. The use of the Nancy Histological Index (Line 165) mentioned at the end of this section stated to be ‘assigned for the highest degree of inflammation per case’ requires some explanation. Clarification would be useful as the NHI rating scale was used to determine histological disease activity in the study (Table 2), but with minimal explanation of the rating scale in the legend.

(3) Discussion (Line 272) The authors state: ‘However, higher disease activity was not identified as an independent predictor of higher NfL values in the present cohort, presumably due to small sample size’. Instead of just asserting this null result, the authors might adopt a more cautious phrasing by adding wording such as….in addition to limitation in sample size, the absence of an independent association between disease activity and NfL values may also reflect measurement sensitivity, or the complex relationship between systemic inflammation and neuroaxonal injury.

**Do you want your identity to be public for this peer review?** For information about this choice, including consent withdrawal, please see our Privacy Policy

Reviewer #2: **Yes:** Oliver Ingvar Wagner

Reviewer #3: No

---

## [Author Response · Author response to Decision Letter 2]

9 Dec 2025

Dear Editors, dear Reviewers,

Thank you very much for your kind suggestions.

In the following we respond to your comments point by point.

Response to Reviewer #2:

Question 1:

The authors provided a thorough revision addressing the reviewers' main concerns. Nevertheless, lack of neurocognitive testing and the relatively small sample size without an a priori power calculation is still an issue. While these are acknowledged in the discussion, they limit the strength of conclusions. Yet, the study is overall technically sound, the data support the main conclusion that serum neurofilament-light chain (NfL) levels show age-dependency but no significant difference between inflammatory bowel disease (IBD) patients and controls, although Crohn's disease showed a distinct age-dependent pattern. The conclusions are appropriately cautious given the limitations.

Minor points:

1) I recommend to emphasize the pilot nature of this study clearly and suggest future larger, longitudinal studies with cognitive testing and multimodal biomarkers.

Reply 1:

Thank you for this remark. We adapted title, abstract, and conclusion section, accordingly, now emphasizing the pilot nature of this study.

Question 2:

2) I suggest adding a Supplementary Information showing other statistical values besides p value such as t and df values for t-test, df, P value for ANOVA, H and df for Kruskal Wallis etc. Violin box plot showing single data points shall be provided and max/min should show the horizontal termination bar (similar to STD error bars, see also https://www.nature.com/articles/s41598-023-29704-8/figures/1). In addition. wildcards should show significance directly in graphs (Fig. 1B).

Reply 2:

Thank you for this important remark. We added comprehensive test statistics as supplementary material. Furthermore, we adapted Figure 1B now depicting a violin box plot. We ask the reviewer to not include “significance” in form of p-values in the graph, since three distinct statistical tests were performed to identify group differences.

Question 3:

3) The author sequence is a bit unusual. It says “equal contribution” for the two last authors though this is usually done for first authors. Also, the last author is commonly the PI receiving funds which is here not the case. I suggest to clarify this.

Reply 3:

This work was conducted in the context of a biobanking study for which Katja Steger, Moritz Middelhoff, and David Schult-Hannemann received funding. This pilot study itself was initiated by the first author, the study received no dedicated funding, and both senior authors—David Schult-Hannemann from the Department of Internal Medicine II and Paul Lingor from the Department of Neurology—jointly supervised the study. We hope that clarifies the ambiguities.

Response to Reviewer #3:

Question 1:

(1) In my view, the authors have adequately addressed the concerns of the prior reviewer. Overall, the manuscript is well constructed, and the conclusions largely are justified by the experimental data presented. I have two minor comments for the authors for their consideration:

(2) Methods. Data Acquisition and Classification. This section (with regard to histological findings) is confusing and could be structured for clearer readability. The use of the Nancy Histological Index (Line 165) mentioned at the end of this section stated to be ‘assigned for the highest degree of inflammation per case’ requires some explanation. Clarification would be useful as the NHI rating scale was used to determine histological disease activity in the study (Table 2), but with minimal explanation of the rating scale in the legend.

Reply 1:

We thank the reviewer for this important comment and have clarified our manuscript accordingly. We added further information about the Nancy index and clarified that, if patients presented with several different grades of histological inflammation, i.e., areas with low inflammatory activity and areas with high inflammatory activity (in one slide or in several biopsies), the NHI was always based on the area with the highest inflammatory activity. This corresponds to routine pathological diagnostics. We have made the following changes, which we hope will clarify any ambiguities:

Biopsy samples were processed routinely and HE staining was performed according to standard protocols. Experienced pathologists performed histologic evaluation and diagnosis of IBD was made in accordance with valid recommendations [34] together with clinical information. Each biopsy site was assessed separately. Histologically, chronic inflammation was defined by a mononuclear inflammatory infiltrate. Active inflammation was present with an infiltration of neutrophil granulocytes, in the presence of crypt abscesses, or cryptitis. If there were features of both, chronic and active inflammation, the case was classified as active-chronic inflammation. Degree of inflammation was classified as low, moderate, or high grade, based on the density of the immune infiltrate. Likewise, crypt architectural disorder was classified in three grades (low, moderate, or high) based on the extent and severity of crypt atrophy, crypt shortening, and crypt branching. Each histopathological report was systematically evaluated, and the findings of the individual biopsy positions were summarized for each individual case. When multiple different grades of inflammation or crypt architectural disorder were present, for example low or high inflammatory activity and crypt architectural disorder in a patient, the highest grade per case was noted. In addition, The Nancy Index was employed to assess and grade the histological severity of inflammation. The Nancy Index uses a five-level scale from 0 to 4, with 0 indicating no relevant histological disease activity and 1 to 4 representing increasing degrees of inflammation and tissue damage, up to grade 4, which reflects severe activity such as ulcers or erosions. The Nancy Index was always based on the highest degree of inflammation. [35]

Question 2:

(3) Discussion (Line 272) The authors state: ‘However, higher disease activity was not identified as an independent predictor of higher NfL values in the present cohort, presumably due to small sample size’. Instead of just asserting this null result, the authors might adopt a more cautious phrasing by adding wording such as….in addition to limitation in sample size, the absence of an independent association between disease activity and NfL values may also reflect measurement sensitivity, or the complex relationship between systemic inflammation and neuroaxonal injury.

Reply 2: Thank you for this remark, we adapted this section to:

In consideration of the small sample size, the absence of an independent association between disease activity and NfL values may also reflect measurement sensitivity, or the complex relationship between systemic inflammation and neuroaxonal injury.

---

## [Editor Report · Decision Letter 2]

18 Dec 2025

Retrospective analysis of neurofilament-light chain in patients with inflammatory bowel disease - a pilot study

PONE-D-25-29221R2

Dear Dr. Wolff,

We’re pleased to inform you that your manuscript has been judged scientifically suitable for publication and will be formally accepted for publication once it meets all outstanding technical requirements.

Kind regards,

Andrea Calcagno

Academic Editor

PLOS One

Additional Editor Comments (optional):

Thanks for addressing the issues raised by the reviewers: I believe the manuscript can now be accepted for publication
---

## [Editor Report · Acceptance letter]

PONE-D-25-29221R2

PLOS One

Dear Dr. Wolff,

I'm pleased to inform you that your manuscript has been deemed suitable for publication in PLOS One. Congratulations! Your manuscript is now being handed over to our production team.

Kind regards,

on behalf of

Prof. Andrea Calcagno

Academic Editor

PLOS One